# Onco-Pathogen Mediated Cancer Progression and Associated Signaling Pathways in Cancer Development

**DOI:** 10.3390/pathogens12060770

**Published:** 2023-05-28

**Authors:** Sandra Kannampuzha, Abilash Valsala Gopalakrishnan, Hafiza Padinharayil, Reema Rose Alappat, Kavya V. Anilkumar, Alex George, Abhijit Dey, Balachandar Vellingiri, Harishkumar Madhyastha, Raja Ganesan, Thiyagarajan Ramesh, Rama Jayaraj, D. S. Prabakaran

**Affiliations:** 1Department of Biomedical Sciences, School of Biosciences and Technology, Vellore Institute of Technology (VIT), Vellore 632014, India; sandrajacobkannampuzha@gmail.com; 2Jubilee Centre for Medical Research, Jubilee Mission Medical College and Research Institute, Thrissur 680596, India; hafizaaficvd@gmail.com (H.P.); reemaalappat007@gmail.com (R.R.A.); vkavyaanilkumar0@gmail.com (K.V.A.); alexgeorge@jmmc.ac.in (A.G.); 3Post Graduate and Research Department of Zoology, Maharajas College, Ernakulam 682011, India; 4Department of Life Sciences, Presidency University, Kolkata 700073, India; abhijit.dbs@presiuniv.ac.in; 5Stem Cell and Regenerative Medicine/Translational Research, Department of Zoology, School of Basic Sciences, Central University of Punjab (CUPB), Bathinda 151401, India; balachandar.vellingiri@cup.edu.in; 6Department of Cardiovascular Physiology, Faculty of Medicine, University of Miyazaki, Miyazaki 889-1692, Japan; hkumar@med.miyazaki-u.ac.jp; 7Institute for Liver and Digestive Diseases, College of Medicine, Hallym University, Chuncheon 24252, Republic of Korea; vraja.ganesan@gmail.com; 8Department of Basic Medical Sciences, College of Medicine, Prince Sattam bin Abdulaziz University, P.O. Box 173, Al-Kharj 11942, Saudi Arabia; 9Jindal Institute of Behavioral Sciences (JIBS), Jindal Global Institution of Eminence Deemed to Be University, Sonipat 131001, India; jramamoorthi@gmail.com; 10Director of Clinical Sciences, Northern Territory Institute of Research and Training, Darwin, NT 0909, Australia; 11Department of Radiation Oncology, College of Medicine, Chungbuk National University, Chungdae-ro 1, Seowon-gu, Cheongju 28644, Republic of Korea; 12Department of Biotechnology, Ayya Nadar Janaki Ammal College, Srivilliputhur Main Road, Sivakasi 626124, India

**Keywords:** pathogens, viruses, bacteria, infections, cancer

## Abstract

Infection with viruses, bacteria, and parasites are thought to be the underlying cause of about 8–17% of the world’s cancer burden, i.e., approximately one in every five malignancies globally is caused by an infectious pathogen. Oncogenesis is thought to be aided by eleven major pathogens. It is crucial to identify microorganisms that potentially act as human carcinogens and to understand how exposure to such pathogens occur as well as the following carcinogenic pathways they induce. Gaining knowledge in this field will give important suggestions for effective pathogen-driven cancer care, control, and, ultimately, prevention. This review will mainly focus on the major onco-pathogens and the types of cancer caused by them. It will also discuss the major pathways which, when altered, lead to the progression of these cancers.

## 1. Introduction

Cancer is one of the leading causes of mortality globally. The International Agency for Research on Cancer (IARC) classifies viruses, bacteria, and parasites as Category I human carcinogens since they are linked to human malignancies [1]. Pathogens are known to infect a huge number of people. Different alterations brought on by infection, including chronic inflammation and other modifications, lead to oxidative stress, which eventually results in cancer initiation. Viruses may trigger cancer by interacting with specific proteins, thriving when the immune system is compromised and invading growing cells. Unlike other viruses, human tumor-causing viruses are unique as they infect yet do not destroy their host cells. This permits human tumor viruses to infect the cells persistently. Since oncogenic pathogens are characterized by chronic infection, it provides a window of possibility for preventing cancer by addressing the pathogen prior to cancerous development [2]. More studies are required for a better understanding of the pathogenesis followed by these infections.

Infectious organisms significantly influence cancer incidence worldwide, and it is vital to recognize that oncogenesis is a rare result of infection and a divergence from the pathogens’ typical life cycle. When pathogen-induced oncogenesis occurs, it typically happens many decades following the first infection [3]. This lag shows that other procedures are necessary in addition to pathogen infection. There are currently seven oncogenic pathogens that significantly result in the progression of carcinogenesis. Epstein–Barr virus (EBV), human high-risk papillomaviruses (HPV), Kaposi’s sarcoma-associated herpesvirus (KSHV), hepatitis B virus (HBV), hepatitis C virus (HCV), Merkel cell polyomavirus (MCPV), and Human T-cell Lymphotropic virus type 1 (HTLV) are some of these major pathogens that cause cancer [4].

## 2. Pathogen-Associated Cancer

### 2.1. Incidence of Pathogen-Mediated Cancer

Pathogen-based infections are believed to be accountable for approximately 15% of cancer incidence rates worldwide, and this figures rises to up to 20% in underdeveloped countries [5]. With the development of new technology permitting genetic identification, these figures are extremely likely to rise further [6]. The International Agency for Research on Cancer has conducted periodic assessments of the cancer burden caused by carcinogenic infections [7]. It indicated that high-risk HPV, Helicobacter pylori, HPV, and HCV were found to be the four most significant viral agents for cancer incidence, accounting for over 90% of infection-related malignancies globally. In 2018, pathogens were responsible for 22 million new cancer cases, accounting for 13% of all cancer cases [7]. Men and women shared the same overall burden of infection-related cancer, but the range of potential pathogens and cancer differed by sex [8].

### 2.2. Major Cancer Types Caused by Pathogens

#### 2.2.1. Hepatocellular Carcinoma (HCC)

The majority of instances of HCC are associated with cirrhosis caused by chronic hepatitis B or C virus infection. The major pathophysiology for HBV oncogenesis is the incorporation of the hepatitis B viral genome into the host DNA. In 60% of HCC cases, viral genome insertion in telomerase reverse transcriptase (TERT) promoter regions of the human genome results in the formation of mutation [9]. In individuals who have been infected with the hepatitis B virus, HCC can also develop in the absence of cirrhosis. Cirrhosis is seen in over 80% of HBV-related HCC cases [10]. Elevated blood HBV DNA levels are a substantial risk factor for HCC in HBV patients. HCV is an RNA virus that does not incorporate into the host genome in the same way as HBV; therefore, the risk of developing HCC is mostly restricted to those with either cirrhosis or chronic disease, along with bridging fibrosis. There have been fewer instances of HCV-related HCC in individuals who do not have cirrhosis. HCV is responsible for 20% of all HCC cases diagnosed worldwide. Viral co-infection with the hepatitis B virus is associated with a higher risk of HCC [11,12].

#### 2.2.2. Adult T-Cell Leukemia

Adult T-cell leukemia is T-cell malignancy caused by infection of the human T-cell leukemia virus type I (HTLV-1). The association between this adult T-cell leukemia and the HTLV-1 virus was observed as all of the patients presenting this type of leukemia had antibodies directed towards fighting the HTLV-1 virus. Another reason was that, most cases reported were from the high incidence rates with HTLV-1 carriers. This particular virus is most endemic in the southern and northern parts of Japan, Africa, and in some parts of America. The two viral oncoproteins, the trans activator protein (tax) and HTLV-1 basic leucine zipper factor (HBZ), are critical in the development and progression of leukemia. Tax protein expression plays a major role in the initiation of neoplastic transformation, while the HBZ protein is involved and expressed in all infected tumorigenic cell lines and is responsible for leukemic cell proliferation [13]. HTLV-1 is typically spread by cell-to-cell contact rather than through cell-free virions. Infected HTLV-1 cells generate viral synapses with uninfected cells. The intercellular adhesion molecule-1 and TAX play critical roles in the development of the virological synapse. Enveloped viral particles can pass across this synapse, thus spreading infection [14].

#### 2.2.3. Merkel Cell Carcinoma

Merkel cell carcinoma (MCC) is a rare kind of skin cancer that commonly recurs and spreads within two to three years of initial diagnosis. MCC has a higher death rate than melanoma (roughly 33%) [15]. Given the growing prevalence of MCC cases and high incidence of MCPyV infection, there is a need to better understand this virus and its carcinogenic potential. Until now, many elements of MCPyV biology and carcinogenic processes are still unknown [16]. The MCPyV virus genome integrates the MCC genome, implying the viral integration happens before tumorigenesis and that MCPyV is a possible etiological factor for this malignancy. Recent research has demonstrated that MCPyV DNA is incorporated in around 80% of MCC cases [17,18]. The virus replicates and remains as an episome inside infected non-malignant cells during long-term chronic infection. Viral DNA found in MCCs is usually inserted into the cell genome. Viral incorporation happens at random places in the genome, though most typically on chromosome 5 [18]. The evolutionary methods that allow MCPyV to infect a significant group of people asymptomatically, frequently, and for a considerable amount of time are still unknown. The fortunate discovery that human dermal fibroblasts sustain MCPyV infection allowed us to recently characterize the MCPyV infectious cycle and the repercussions for the host cell [19].

#### 2.2.4. Cervical, Head and Neck, and Anogenital Tract Carcinomas

Head and neck squamous cell carcinomas (HNSCCs) account for 4.8% of all malignancies, and are linked to a comparable percentage of cancer deaths globally. Human papillomavirus is generally known as the leading source of cervical cancer, although in recent years, the definite involvement of this infection in other cancers has emerged [20]. HPV infections are spread orally or via the anogenital region. The increased transmission of HPV through oral transmission results in the head and neck region being infected [21]. It was observed that around 3 to 5% of HPV infections occurring in the cervix result in the transforming infections based on the cell origin [22]. Around 90 different types of HPV have been reported. Among them, 16 and 18 are most commonly associated with oncogenic progression [23].

## 3. Types of Onco-Pathogens

Pathogen-based infections cause the development of oncogenes, stimulate DNA damage and inflammation from a protracted infection, and inhibit the host’s immune system [24]. Cancers have been linked to viruses such as CMV, HHV-8, HPV, HBV, HCV, and EBV. According to another study, a modified bacteriophage population may promote carcinogenesis by permitting the growth of opportunistic, cancer-causing bacteria in gastrointestinal biofilms [25]. Fibiger initially brought attention to the connection between parasites and cancer in 1926. He discovered that Spiroptera-infected mice gradually developed stomach cancer. Several researches later refuted this. Nonetheless, this stoked interest in protozoa’s potential connection to cancer and resulted in the discovery of novel theories regarding protozoa’s function in the development and spread of cancer [26].

### 3.1. Direct and Indirect Onco-Pathogens

Pathogens are typically grouped into direct and indirect carcinogens, although the oncogenic mechanisms they employ are quite diverse (4). In direct pathogens, such as HTLV-1, HPV, MCPyV, EBV, and KSVH, all cancerous cells contain crucial segments of the viral genome, which prevent apoptosis and interrupt the cell cycle and cell immortalization. Indirect pathogens, including *H. pylori*, *S. haematobium*, HBV, HCV, *O. viverrini*, and *C. sinensis* do not promote the production of oncogenes; however continuous infection can result in a chronic inflammatory state.

Bacteriome dysbiosis can cause cancer in organs that lack their own bacteriomes. This can be seen in hepatocellular cancer (HCC), colorectal cancer (CRC), lung cancer (LC), and gastric cancer (GC). By inducing epithelial damage that culminates in inflammation and later carcinogenesis, Helicobacter pylori has the capacity to directly cause stomach cancer [27]. CRC in humans caused by *Escherichia coli*, *Bacteriodes fragilis*, and *Fusobacterium nucleatum* infections can activate NF-kB signaling, triggering the Wnt pathway. This pathway leads to the production of genotoxins that can cause cancer. The binding of bacterial products, including MAMPs and lipopolysaccharides (LPS) to TLRs on immune cells stimulates the oncogene Kras—a key pancreatic cancer inducer [28].

Several diseases, including periodontal disease, HIV, UTIs, and inflammatory bowel disease, have already been linked to modifications or additions to the human virome, which are mostly associated with human viruses and bacteriophages [29]. In terms of genetic modifications, effects on cellular networks, chronic inflammation, and the possible effects that bacteriophages could have on the bacterial community, all of these viruses interact with carcinogenesis in a unique way [30]. Further study is required to determine how interactions between the viral community and the bacteriome may promote or prevent carcinogenesis.

### 3.2. Oncogenic Viruses

Six human viruses are strongly linked to cancer, and around 15% of all human malignancies are caused by viruses [7], including RNA viruses (HTLV-1, HCV) and DNA viruses (MCPyV, HR-HPV, EBV/HHV-4, KSHV/HHV-8, and HBV) [31]. Viral oncoproteins allow cells to escape immune destruction, maintain multiplication and immortalization, cause mutations and genetic changes, enhance chronic inflammation, and promote metastasis and angiogenesis. Viral oncoproteins also cause imbalances in cellular energetics [32]. Oncoviruses can affect cellular gene expression by modifying host DNA methylation, triggering chromatin reorganization, expressing non-coding RNAs generated by the virus, and affecting cellular non-coding RNAomics [33].

#### 3.2.1. Epstein–Barr Virus (EBV)

EBV is a DNA virus typically linked to Burkitt’s lymphoma. It is an encapsulated virus that spreads via body fluids, such as saliva and genital secretions [24]. Infection by EBV has also been linked to the development of lymphoma and GC [34]. Alterations in EBV induce GC through the methylation of the host genome. With the aid of B cells, an oral virus enters stomach epithelial cells and attaches to them via the host cell receptors before becoming an episome [35]. After the establishment of latent infection, it promotes cell growth and metastasis and disturbs the host’s response to DNA damage through viral the EBER-1 and -2 non-coding RNAs and EBNA1, all of which induce ROS build-up [36]. By methylating CpG islands, the EBV LMP2A causes epigenetic alterations to the host genome, inactivating tumor suppressor genes, including PTEN and tumor-associated antigens [37]. In fact, EBVaGC induces the hypermethylation of the promoters of the CDH1, p15, p16INK4a, p14ARF, and p73 tumor suppressor genes [38]. Viral genome methylation not only alters the host genome but also aids in pathogen evasion from the host immune system. Additional features of EBV-associated cancers include distinctive DNA methylation patterns, commonly known as the CpG-island methylator phenotype (CIMP), which are seen in nasopharyngeal carcinoma, Hodgkin’s lymphoma, and gastric cancer. These epigenetic alterations are brought on by LMP-mediated DNMT1 overexpression [39]. EBV can induce soft tissue neoplasms (smooth muscle tumors). Initially, these was observed in individuals who suffer from immunosuppression [40]. Occasionally, iatrogenic immunosuppression for an autoimmune illness leads to EBV+SMT. AKT/mTOR pathway activation, and MYC overexpression are promising therapeutic targets [40].

#### 3.2.2. Kaposi’s Sarcoma Virus

Kaposi’s sarcoma virus was identified as the causative agent for Kaposi’s Sarcoma (KS) [41]. It can develop on the lining of the nose, skin, mouth, lymph nodes, or other important organs. The lesions, which are frequently purple in color and composed of lymphatic endothelial cells or their precursors, might progress to become clonal metastases of spindle cells in more advanced stages [42]. The virus encodes proteins that are similar to those produced by its human host, including cyclin, viral FLICE inhibitory protein (vFLIP), BCL-2), IL-6, viral FLICE inhibitory protein (vFLIP), IRFs, and chemokines. Viral vFLIP and cyclin encourage the growth of tumor cells that have been infected while the virus is latent. KSHV is associated with both KS and lymphatic system tumors, as well as Castleman’s disease and primary effusion lymphoma [43]. This virus enters the host through saliva and spreads to a range of cells, including fibroblasts, endothelial, and epithelial cells, and it can also affect immune system cells, including monocytes, dendritic cells, and B cells [44]. In the nucleus of the target cells, the virus becomes exposed along with its genome to generate an episome, where it will either go dormant or undergo cycles of lytic reactivation [45]. In experimental settings, KSHV oncogenic proteins have been found to inhibit apoptosis; however, for the establishment of carcinogenesis, additional co-factors must be present, such as co-infection with the HIV virus or the host’s consumption of immunosuppressive medications. Tat, a transcriptional trans-activator that is encoded by HIV-1, promotes KSHV infectivity and causes CD4+ T cells to undergo apoptosis. Nef, a different HIV protein, controls the AKT signaling pathway, increasing the levels of vIL-6 and other cytokines to promote angiogenesis, KSHV oncogenesis, and the reactivation of the KSHV life cycle via JAK/STAT signaling [24].

#### 3.2.3. Human T-Cell Lymphoma Virus 1

Human T-cell lymphotropic virus type 1 (HTLV-1) was the first oncogenic retrovirus discovered in humans, and it was discovered in the early 1980s by two distinct research teams in Japan and the USA. The Delta retrovirus genus and Retroviridae family both contain the enveloped complex retrovirus known as HTLV-1. This genus contains three more HTLV members: HTLV-2, -3, and -4. Chronic lymphoma and acute ATLL are associated with a type of tumor of CD4+CD25+ T cells that is brought on by HTLV-1 [46]. Horizontal transfer of diseased lymphocytes through sexual interaction and vertical transmission from a mother (carrier) to her infant through nursing are two examples of ways that HTLV-1 can spread [47]. The attack of HTLV-1 marks the beginning of viral persistence and reproduction. HTLV-1 binds to its receptor on target cells such as heparin sulfate proteoglycan (HSPG), glucose transporter (GLUT1), and the VEGF-165 receptor neuropilin-1 (NRP-1), the latter of which instructs the infiltration process [48]. The subsequent interaction of GLUT1 with the HSPG/NRP-1 complex results in a fusion. The viral RNA is then transported into the target cells’ cytoplasm by a viral core. During reverse transcription, HTLV-1 integrates its genome into the host genome to form a provirus, and this provirus then uses transcription to make structural regulatory and auxiliary proteins. Then, an immature viral particle is formed by the combination of the viral genomic RNA with the Gag, Env, and Gag-Pol proteins at the plasma membrane. In order to become a mature and contagious viral particle, the budding particle must first be liberated from the cell exterior [49]. The HTLV-1 genome also contains genes that encode the non-structural proteins. Tax and HBZ are essential for controlling viral gene expression. By controlling a number of intracellular signaling pathways, such as the IKK/NF-B, DNA damage reparation, and innate immune signaling pathways (RIG-I/MDA5-dependent, TLR-independent, TRIF-dependent TLR pathways), and the recently discovered cGAS-STING pathway, Tax-1 has a central role in tumorigeneses and contributes to ATL [49]. HBZ controls cell growth by forming heterodimers with host factors such as CCAAT/enhancer binding protein (C/EBP) and activating transcription factor 3 (ATF3). ATL cell motility and proliferation are promoted by HBZ through increasing the noncanonical Wnt5a production. Recent studies have revealed that HBZ promotes the mTOR pathway by inhibiting growth arrest and the stress-induced GADD family member GADD34, which inhibits the mTOR pathway [50].

#### 3.2.4. Human Papilloma Virus

According to experts, the most common sexually transmitted disease (STD) is HPV. Despite the fact that infections are asymptomatic 90% of the time and go away in 1 or 2 years, HPV is nonetheless linked to more than 500,000 new cases of cancer each year globally [51]. Due to the distinctive genetic traits of high-risk HPV types, HPV-related lesions can occasionally develop into malignant tumors. Alpha-papillomavirus types 51 (clade A5); 56, 66 (clade A6); 18, 45, 59 (clade A7); 16, 31, 52, 58 (clade A9) are classified as high-risk HPVs by the IARC monograph [52]. Specifically, the HPV types 16 and 18 cause cervical cancer through the activation of oncogenes E6 and E7 [39]. Cervical, head and neck, anus, vulva, penis, and oropharynx cancers are all known to be caused by the human papilloma virus (HPV) [53]. The fourth-highest incidence and mortality rate among female malignancies in 2020 was that of cervical cancer [54]. The p16 marker is employed in practice to ascertain if cervical cancer is related to high-risk HPV [55] and also for HPV-associated squamous cell carcinoma (Table 1).

#### 3.2.5. Hepatitis Virus B and C

The hepatitis B and C viruses, identified in 1969 and 1989, respectively, are key risk factors for the third-leading cause of mortality due to the cancer hepatocellular carcinoma (HCC). Given how often HBV integrates into host DNA, insertional mutagenesis appears to be a key carcinogenic process in HBV-related carcinogenesis [72]. When hepatitis is left untreated, ongoing inflammation and liver damage cause cirrhosis, which, in turn, results in HCC [73]. The dysregulation of the p53, TERT, and WNT pathways, mostly caused by mutations in the genes encoding TP53, the TERT promotor, and CTNNB1, respectively, is one of the major driving forces behind HCC [61,62]. Small duct type intrahepatic carcinoma exhibits a biliary character, which is similar to risk of developing HBV [74,75]. HCV unquestionably causes hepatitis, unlike HBV [76]. A number of HCV proteins also exhibit carcinogenic qualities in addition to their indirect effects [77]. Hepatocellular carcinogenesis is therefore prevented by HCV eradiation by direct antiviral agents (DAAs). However, when the virus is eradicated, liver cirrhosis increases the risk of cancer. The discovery of novel therapeutic medications able to stop the progression of sickness into cancer requires a thorough exploration of the molecular mechanisms connecting HCV infection and HCC [4].

#### 3.2.6. Merkel Cell Polyomavirus

Merkel cell carcinoma (MCC) is where MCV was initially discovered and includes five viral genes, including small and large T antigens (sT and LT), VP1, VP2, and VP3 [78]. As is the case with other polyoma viruses, LT plays a critical role in the development of cancer. Merkel cell carcinogenesis likely involves LT alterations as an oncogenic event [79]. Studies conducted in vitro and on animals, as well as the discovery of sT in certain MCC without the presence of LT, suggest that sT may play a more significant role in the oncogenic process than LT, which is necessary to maintain tumor cell development [80]. A diagnostic marker for MCC is an anti-LT antibody called CM2B4, which is created from exon 2 of the LT gene in MCV [81]. Mutations in Rb, TP53, and PIK3CA, as well as L-myc amplification, are other frequent chromosomal alterations in MCC [79]. MCV-negative MCC is more likely to have a higher tumor mutation load and drive mutations due to UV exposure [82]. As indicated by the combined squamous and neuroendocrine carcinoma of the skin that is MCV-negative, cutaneous neuroendocrine carcinoma that is MCV-negative may only be brought on by UV exposure as opposed to MCC that is MCV-positive [82].

### 3.3. Oncogenic Bacterium

The bacterial microbiome, also known as the bacteriome, changes during a person’s lifetime due to environmental and genetic variables related to the host [25]. Dysbiosis, i.e., harmful changes in the bacteriome, can cause cancer because of the growth of oncogenic bacteria and the impact of bacterial metabolites on the host, even in organs without their own bacteriomes [27].

*Helicobacter pylori* causes peptic ulcers. Experts started to speculate that there could be a passing connection between this bacterium and stomach cancer. The WHO and International Agency for Research on Cancer (IARC) originally identified *H. pylori* as a human carcinogen (group I). *H. pylori* is thought to be the primary culprit behind the inflammation of the stomach that leads to peptic ulcer disease (10–20%), gastric mucosal-associated lymphoid tissue (MALT) lymphoma (1%), and distal gastric adenocarcinoma (1–2%) [83].

Moreover, a unique bacteriome that has been linked to pancreatic cancer is hypothesized to facilitate the development of oncogenesis by impairing peritumoral immunity and creating a tumor-promoting milieu [84]. Human CRC has been linked to several bacteria, including *Fusobacterium nucleatum*, *Escherichia coli*, *Streptococcus gallolyticus*, and *Bacteriodes fragilis*.

In an in vivo model, infection with *F. nucleatum* increases the number of tumor cells and attracts myeloid cells that invade the tumor [85]. The epithelial–mesenchymal transition pathway was subsequently activated by the bacterium’s ability to increase the proliferation of healthy human colon cells [86]. Several studies have shown an abundance of *F. nucleatum* in fecal matter as well as in the tumor tissues of CRC patients. In terms of molecular characteristics, *F. nucleatum* is associated with significant microsatellite instability, the CpG island methylator phenotype, and various gene mutations. It is also associated with reduced levels of CD3+ T cells [87]. The cell surface proteins FadA, Fap2, and RadD, all of which are produced by *F. nucleatum*, can drive the host to release inflammatory factors and engage inflammatory cells, creating an environment that is favorable to tumor formation [88].

*Bacteroides fragilis* (*B. fragilis*) is a common causative agent of CRC. It is mainly divided into two classes, wherein enterotoxigenic *B. fragilis* is the most prevalent type that carries the toxin [89]. *B. Fragilis* Toxin (BFT). *B. fragilis* produces biofilm for colonization in the intestinal tract, which can trigger a cascade of inflammatory events that leads to the release of BFT [90]. This causes persistent inflammation of the intestines and tissue injury and plays a critical role in CRC. Several other studies also show that ETBF (Enterotoxigenic *Bacteroides fragilis*) can also induce CRC by triggering the NF-κB or Wnt signaling pathways—enhancing polyamine metabolism, resulting in DNA damage, and stimulating Th17 adaptive immunity [91].

*Streptococcus gallolyticus (S. gallolyticus*) is another causative agent of CRC. CRCs have been identified in 25 to 80% of individuals with *S. gallolyticus* [92]. *S. gallolyticus* is a leading cause of endocarditis (inflammation of the heart’s inner layer). A strong relationship between endocarditis and CRC was established long ago by McCoy and Mason [93]. Studies have shown that *S. gallolyticus* initiates the recruitment of CD11b+ myeloid cells, which promotes the progression of colitis-associated cancer. Given the effect of increasing *S. gallolyticus* on myeloid cells and immune suppression, it appears that *S. gallolyticus* and tumor cells interact to drive their survival and invasion during carcinogenesis [94]. A better understanding of the involvement of the bacteriomes in cancer and their mechanism in inducing the cancer can aid in the creation of cutting-edge treatments and diagnostic methods that may help physicians treat cancer patients.

### 3.4. Oncogenic Parasites

Parasitic diseases caused by protozoans or helminths play an important role in favoring carcinogenesis. They act as inducers or promoters of cancer; however, their regulatory effect on tumorigenesis is much less studied. In 37 nations, there has been evidence of a link between adult brain cancer and *Toxoplasma gondii* prevalence, while, in France, there has been evidence of a link between increased brain cancer mortality and a higher seroprevalence of *Toxoplasma gondii* [95]. It is also been proposed that there is a rise in the prevalence of rectal and nasopharyngeal cancers in toxoplasmosis-infected people [96]. *Trichomonas vaginalis* is connected to two cancers of the reproductive system, including prostate cancer in men and cervical cancer in women [97]. Infected cells have a *T. vaginalis* protein that is similar to the human macrophage inhibitory factor (MIF). The TvMIF protein opens pathways that are important in inflammation and cell growth. TvMIF silences the BAD protein in order to activate the Akt pathway [98].

Two species of liver flukes, *Opisthorchis viverrini* and *Clonorchis sinensis*, are the pathogens responsible for food contamination [99]. The IARC classed these parasites as category I human carcinogens in 2012. The urogenital schistosomiasis disease, which is brought on by *Schistosoma haematobium* parasites, is linked to cancer throughout the Middle East and Africa [71]. This infection causes hematuria, a disease linked to the emergence of bladder cancer, which is characterized by the presence of blood in urine and persistent inflammation [100]. In animal models, H03-H-IPSE, an ortholog of interleukin-4-inducing principle (IPSE) protein, causes bladder angiogenesis, induces urothelial cell proliferation, and permits *S. haematobium* eggs to evade the host’s immune system. The eggs deposited on the wall of the bladder damages and irritates the lumen of the bladder, increasing the likelihood of bladder cancer [101]. *S. haematobium* was categorized as a category I definite biological carcinogenic agent by the IARC in 2012 due to its tendency to generate genetic and epigenetic alterations that result in cell hyperplasia and cancer [102]. A key technique for avoiding the majority of tumors is the elimination of infections linked to tumors via vaccination or treatment.

## 4. Mechanistic Action of Carcinogenesis by Pathogens

To protect from cellular modifications that may take place when cells are affected by oncogenic pathogens, tumor suppressor pathways must be activated. Cellular responses that follow, such as cell cycle arrest, senescence, and apoptosis, may halt cancer from developing, stop the reproduction of pathogens, and repair DNA damage. The two primary tumor suppressor mechanisms, which strictly regulate cell cycle progression, stimulate DNA damage, and induce apoptosis following irreversible cell damage, are centrally controlled by the cellular tumor antigens retinoblastoma (pRB) and p53 [103]. Oncoproteins that deregulate pRB and p53 pathways are encoded by almost all onco-pathogens, although their underlying mechanisms vary [2]. Viral oncoproteins cause the inactivation, degradation, suppression, or detachment of pRB and p53 from their appropriate functional partners—inhibiting their functioning [104].

Viral spread may be aided by the deregulation of pRB and p53’s tumor suppressor functions. For instance, the oncoproteins programmed by large oncogenic herpesviruses (EBV or KSHV) and small DNA oncogenic viruses (HPV) can inhibit the activity of pRB and p53 to cause the cell to enter into S phase, i.e., the process of DNA synthesis, allowing the virus to gain entry to the cellular replication processes and nucleotides for viral DNA generation [104]. Moreover, HTLV-1 oncoproteins trans activator from basic zipper factor (HBZ) and X-gene region (Tax) both have the ability to suppress p53 activity via a variety of methods that encourage carcinogenesis [105]. In HBV-mediated hepatocellular carcinoma, the p53 and pRB mechanisms are typically deregulated; the viral HBV-X protein (HBx) creates a link with p53 and suppresses the latter’s transcription factor binding and DNA binding activities [106].

One of the main host defenses against viral infection is the destruction of virally infected cells by apoptosis. Hence, viral reproduction and spread are permitted prior to the death of the host cell when oncogenic viruses suppress apoptotic signaling. To circumvent human defenses against infection and maintain a sustained infection, almost all oncogenic viruses have developed sophisticated apoptosis escape tactics that target pRB and p53 pathways [3,107]. Host cells are at risk of genomic instability and chromosomal abnormalities when viruses target apoptotic pathways and cell cycle check points. Cancer may develop as a result of cells in this modified environment, acquiring several genetic abnormalities and modified signaling pathways (Figure 1).

### 4.1. Reprogrammed Signaling Pathways by Onco-Pathogens

#### 4.1.1. Mitogen-Activated-Protein Kinase (MAPK) Pathway

The transcription of genes that drive antiviral immune response and cell proliferation is modulated by mitogen-activated protein kinase (MAPK) pathways. It aids in the life cycle and transmission of oncogenic viruses, including HPV, MCPyV, and HCV, by promoting viral packaging and release. For instance, the cytosolic phospholipase A2 (PLA2G4A) activity based on the MAPK pathway contributes to the development of infectious HCV components [108]. In the lack of PLA2G4A35, arachidonic acid, the fragmentation product of lipolysis facilitated by PLA2G4A, restores the generation of infectious HCV particles. This shows that the lipid sheath of developing viral particles may effectively incorporate core proteins. According to enhanced HPV virion generation following stimulation of extracellular signal-regulated kinase (ERK)-1 and ERK2 in HPV-affected cells [109], MAPK signaling also increases the formation of non-enveloped viruses. This observation is supported by the fact that the cancer treatment trametinib severely restricts MCPyV infection by preventing MCPyV replication and transcription in infected cells, indicating that the MAPK cascade must be activated to promote these processes in the MCPyV life cycle. Nevertheless, it is uncertain if MAPK signaling also contributes to the growth of Merkel cell carcinoma (MCC)—linked to MCPyV.

Oncogenic viruses commonly change the MAPK pathways to promote the growth of the host cells, but this may also lead to the creation of invasive cells that help the cells spread and metastasize. When EBV infection shifts from a latent to lytic phase, the p38 MAPK axis is crucial for inhibiting apoptosis and causing viral reactivation. During the latent-to-lytic transition, EBV LMP1 is activated, and this avoids apoptosis and promotes reactivation [110]. The ERK/MAPK pathway is activated when LMP1 is expressed in epithelial cells, which encourages cell invasion and metastasis. In this approach, LMP1 may support cell invasion in nasopharyngeal carcinomas linked to EBV [111].

The molecular mechanism by which KSHV activates MAPK cascades is more known than it is in the case of EBV infection [112]. The KSHV kaposin B protein interacts with and triggers MK2 kinase (MK2K), an activator of p38 MAPK signaling cascades, which, in turn, stabilizes mRNAs for pro-survival and pro-inflammatory cytokines. In KSHV-based oncogenesis, enhanced cytokine secretion could encourage the development and persistence of tumor cells. Another example is the way the HBV HBx protein activates the ERK cascade and results in the development of FOXM1—a crucial regulator of tumor metastasis. FOXM1 leads to HBV-mediated hepatocarcinogenesis by upregulating the transcription of RHOC, ROCK1, and MMP7, all of which promote the metastasis of hepatoma cells [113]. Increased invasiveness, deregulated cell division, and enhanced cytokine secretion via hyper MAPK signaling may create the ideal environment for virus propagation, but they also contribute to cancer pathogenesis and resistance to therapy.

#### 4.1.2. β-Catenin (βcat) Pathway

Many physiological functions, including growth regulation, stem cell regeneration, embryogenesis, and tissue determination, are regulated by the WNT/β-catenin signaling pathway. Several development-based diseases, including cancer, may be attributed to the upregulation of the downstream activators of WNT/β-catenin signaling [114]. β-Catenin, for instance, may be stabilized by EBV LMP2A and KSHV LANA proteins, which potentiates downstream genes, including MYC and CCND1, to accelerate cell proliferation and support tumorigenesis [115].

HBx and Hepatitis B surface antigen (HBsAg) are two proteins that HBV encodes and that abnormally stimulate WNT/β-catenin signaling. Inhibitors of the pathway are silenced or their essential elements, such as β-catenin, are upregulated and stabilized by HBsAg and HBx. Collectively, these processes activate target genes that abnormally promote cell proliferation, which eventually aids in the formation of HCC59. Similar to this, the ongoing production of HTLV-1 HBZ in T cell leukemia cells caused by the virus disrupts the WNT signaling cascade, promoting proliferation and migration [116].

While the function of WNT/β-catenin in oncogenic virus infection may provide significant hints, its significance in other viral malignancies is less known. For instance, the induction of downstream MMP genes promotes MCPyV infection by damaging the host cells’ extracellular matrix (ECM) [117]. Stimulation of the pathway also promotes MCPyV infection. The stimulation of WNT/β-catenin signaling and the production of MMPs may result from skin damage brought on by UV and ionizing radiation, as well as ageing processes. According to this, these important risk factors for MCC that are associated with MCPyV increase viral infection and, as a result, tumor formation via MMP induction.

Furthermore, Wnt genes and Snail are directly inhibited by the miRNA-34 family, a transcription target of p53, which in turn prevents p53 from mediating Wnt signaling and endothelial-to-mesenchymal transition (EMT) mechanisms [118]. The potential advancement of cancer is associated with the stimulation of the CXCR4/CXCL12 axis. EMT and miR-125b are both simultaneously upregulated in human colorectal carcinoma (CRC) cells when the CXCR4/CXCL12 axis is activated. MiR-125b upregulation strongly induces EMT and cancer invasion and promotes CXCR4 production. Notably, the adenomatous polyposis (APC) gene is the focus of the reciprocal positive feedback loop among CXCR4 and miR-125b, which further stimulates Wnt/β-catenin signaling [119]. A nucleolar protein called G protein nucleolar (GNL)-3 is abundantly produced in stem cells, progenitor cells, and several kinds of cancer cells. GNL3 is vital for maintaining dryness, telomere integrity, and genomic stability, as well as for cell cycle control, differentiation inhibition, cell proliferation, and ribosome biogenesis.

Through the activation of the Wnt/β-catenin signaling cascade, GNL3 induces EMT in colon cancer [120]. β-arrestin1 promotes the growth of a variety of tumor forms. By turning on the Wnt/β-catenin signals to control the EMT process, β-arrestin1 has the capacity to encourage the movement of CRC cells [121]. A vesicle transporter protein called GOLT1B is involved in the transfer of cytoplasmic proteins. Overexpression of GOLT1B may boost DVL2 levels and plasma membrane trafficking, which in turn activates the Wnt/β-catenin cascade and raises nuclear -catenin levels, causing EMT [87]. A TRIP13 small molecule inhibitor called DCZ0415 prevents EMT and metastasis in CRC by deactivating the Wnt/β-catenin pathway and the FGFR4/STAT3 axis [122].

#### 4.1.3. PI3K–AKT–mTOR Signaling

Phosphatidylinositol 3-kinase/mechanistic target of rapamycin (PI3K/mTOR), which controls macromolecule synthesis and metabolism in response to nutritional availability, is a crucial eukaryotic nutrition-sensing system. By synchronizing growth stimulus and modulating downstream effectors such as AKT and mTOR, it plays a crucial role in the control of cell cycle progression, proliferation, growth, survival, longevity, and quiescence. The proliferation and survival of cancer cells may be caused by PI3K axis deregulation, which can also impair normal cellular growth control [122]. In order to activate this process in the absence of the required nutrition levels and growth factors, several oncogenic viruses, such as EBV, HTLV-1, HPV, MCPyV, and KSHV, have developed mechanisms. By encouraging cell growth and suppressing autophagy, which may obstruct viral replication, the activation of the PI3K/AKT/mTOR signaling may be advantageous for pathogenic infection [123,124].

The HPV example is one that has received the most attention since each of the viral oncoproteins E5, E6, and E7 impact the pathway and encourage cell proliferation, favoring infected cells for tumor development and progression [125]. AKT phosphorylation and PI3K/AKT cascade are both induced by the EBV latent membrane protein (LMP)-2A [126]. Because of this, LMP2A-expressing B cells have a selective advantage during the emergence of EBV-based malignancies. This leads to anti-apoptotic activity, which limits the clearance of injured cells.

The prevention of epithelial cell differentiation via the LMP2A-based activation of the PI3K/AKT pathway in EBV-infected cells suggests that the same mechanism underlies the emergence of EBV-based lymphomas and carcinomas [127]. Long latent phase is promoted by HTLV-1 modulating AKT in CD4+ T cells [128]. Forkhead box protein (FOXO)-3, which depletes CD4+ T cells by inducing anti-proliferative and pro-apoptotic target genes, is inactivated by the HTLV-1 Tax oncoprotein when the AKT pathway is activated. As a result, FOXO3 restriction encourages CD4+ T cell survival and expansion, maintaining their ability to disseminate infectious HTLV-1 particles. At the early stages of HTLV-1 pathogenesis, this activity of the Tax protein allows for the long-term persistence of infected CD4+ T cells.

The finding that the mTOR antagonist rapamycin, in contrast to other immunosuppressants, encourages tumor reduction in transplant patients afflicted with KSHV-based Kaposi sarcoma [129] underlines the significance of mTOR signaling cascade in KSHV activity. Later on, researchers found that the lytic gene KSHV ORF45, which is activated in infected lymphatic endothelial cells, specifically upregulates mTOR signaling [130]. The susceptibility of KSHV-infected cells to rapamycin-based apoptosis may be attributed to the fact that they rely on the mTOR signaling cascade for survival. Inducing sarcoma genesis via the stimulation of AKT phosphorylation only requires the induction of the KSHV G protein-coupled receptor (vGPCR) in a mouse allograft model [131]. The discovery of strong AKT activation in Kaposi sarcoma tissue samples obtained from people with AIDS31 provided evidence supporting the involvement of AKT in the development of human Kaposi sarcoma. In B cells, the KSHV K1 protein stimulates AKT cascade to prevent apoptosis, indicating that this is a strategy to guard against early cell death during KSHV-based oncogenesis [132].

The MCPyV small T oncoprotein targets the PI3K/AKT/mTOR signaling cascade further downstream. It promotes the important mTOR complex 1 target, 4E-BP1, to become hyperphosphorylated, which then results in the hyperactivation of cellular transformation and cap-based protein translation [133]. Through the targeting of mTOR, which normally responds to a network of stimuli comprising amino acid ingestion and environmental stress, each of these biologically distinct viruses causes anabolism.

#### 4.1.4. Notch Signaling

According to the cellular milieu and tissue milieu, changes in the Notch signaling system may either promote or repress tumorigenesis [134]. Notch signaling has been linked to the emergence of breast cancer, chronic lymphocytic leukemia, and B cell malignancies [135]. In contrast, pancreatic cells and skin epithelia may limit tumor growth due to notch signaling. The diverse relationship of Notch signaling with various malignancies is mirrored in a range of ways; therefore, viruses utilize this route in contrast to the pathways examined in the preceding sections, which are essentially elevated in all cancers. Notch signaling was discovered to be a crucial pathway that is targeted by HPV, MCPyV, and EBV oncoproteins, highlighting its significance in viral tumorigenesis [136]. Through the targeting of mTOR, which normally responds to a network of stimuli comprising amino acid ingestion and environmental stress, each of these biologically distinct viruses causes anabolism.

The HPV E6 oncoproteins suppress Notch signaling and aid in the survival of the virus in basal epithelial cells. In order to prevent Notch transcriptional activity, -genus HPV E6 proteins hit Mastermind-like protein (MAML)-1 and a number of other Notch transcription components [137]. Other cutaneous HPVs, including HPV-8, employ a comparable strategy through their E6 proteins to block the Notch-based transcription of the HES1 transcriptional inhibitor [138], preventing keratinocyte differentiation, which has been linked to the role of HPV in furthering cell proliferation and oncogenesis. In order to create a cellular milieu that is favorable for long-term infection, EBV also disrupts Notch signaling [139]. As a result of their competition for the recombining binding protein Suppressor of Hairless (RBP-Jk), the Epstein–Barr nuclear antigen (EBNA)-2 and Notch restrict the transcription of EBV genes associated with the transformation of infected B cells. By inhibiting EBNA2’s ability to promote transcription, continuous Notch signaling in the lymphoid milieu may cause EBV latency [139].

Furthermore, KSHV and HBV stimulate Notch signaling. The neurogenic locus Notch homologue protein (NOTCH)-1, which enhances the production of the HBV HBx protein and encourages the development of HCC cells, may lead to the oncogenic process of HBV-based HCC [140]. In Kaposi sarcoma tumors linked with KSHV, elevated amounts of stimulated Notch proteins are typically found [141]. Viral GPCR, replication and transcription activator (RTA), latency-associated nuclear antigen (LANA), viral FLICE inhibitory protein (vFLIP), viral interleukin-6 (vIL-6), and viral FLIP are KSHV proteins that can induce the activation of core Notch receptors and ligands [142]. Several viral proteins appear to prevent surrounding, uninfected cells from expressing genes associated with the cell cycle, reducing those cells’ proliferation and perhaps providing infected cells with a survival and growth edge during Kaposi sarcoma development. In a process known as EMT, vFLIP and vGPCR-induced Notch cascade stimulation also causes transcriptional remodeling of the infected lymphatic endothelium cells into mesenchymal cells. The spread of the virus is aided by the development and movement of infected cells, which also increases the invasiveness of Kaposi sarcoma [142]. KSHV LANA competitively inhibits the association between an E3 ubiquitin ligase, F-box/WD repeat-containing protein (FBXW)-7, and NOTCH1’s intracellular domain (ICD), limiting ICN53’s proteasomal degradation. In turn, stabilized ICD serves as a proto-oncogene and encourages the growth of tumor cells that are infected by KSHV, increasing virus-based transformation [143]. Notch signaling regulation may result in viral oncogenesis and raises the possibility that physiological circumstances, as well as the kind of infected cell, may affect transformation.

#### 4.1.5. NF-κB Signaling

The nuclear factor-κB (NF-κB) cascade is triggered by inflammatory cytokines and pathogens, which cause the expression of genes involved in a variety of cellular activities, including the inflammatory and innate immune responses [144]. By preventing apoptosis, promoting cell proliferation, and promoting invasiveness, the downstream target gene activation of NF-B in chronic infection and inflammation also accelerates the development of cancer [144]. While NF-κB stimulation plays a role in the proper response to acute viral infections, viruses that attack adaptive immune cells may also employ constitutive NF-κB stimulation to strengthen their host environment. Through triggering downstream target genes of NF-κB, the EBV oncoprotein LMP1 promotes the growth of lymphomas [145]. It does this by imitating constitutively active host tumor necrosis factor receptor (TNFR) and activating the proximal signal transducers of the NF-κB cascade—TNF receptor associated factor (TRAF)-6 and interleukin-1 receptor-associated kinase (IRAK)-1. The multiplication and survival of infected B cells are encouraged by this LMP1-based NF-κB activation [146]. In most KSHV-based primary effusion lymphoma (PEL) cells, NF-κB is also persistently activated [147]. The KSHV vFLIP protein interacts with an antagonist of the NFkB (IKK) complex subunit in these cells, causing a structural shift that makes the IKK complex persistently active [148]. The activation of the NF-κB cascade results from this interaction. When KSHV vFLIP-expressing transgenic mice are exposed to a higher incidence of lymphoma, it is associated with NF-κB -based lymphocyte proliferation [149]. Similarly, the HTLV-1 Tax protein, which is thought to be the main mechanism by which this virus changes T cells, has the ability to activate NF-κB as one of its functions [150]. NF-κB activation brings attention to the inflammatory processes’ ostensibly opposing functions in infection and malignancy.

For its antibacterial effects and preservation of cellular homeostasis, NF-κB is significantly active at the point of infection in response to stimulation by a variety of bacterial species (*F. nucleatum*, *H. pylori*, etc.) [151]. The considerable activation of the NF-κB pathway during the immune response, whether through T or B cells, increases the severity and spread of inflammation. Intrinsic persistent inflammation may lead to tissue destruction, autoimmune diseases, and the emergence of cancer because it increases cellular stresses and accumulates DNA damage. Modifications in the epigenetic and genetic modifications produce a pro-tumorigenic environment at the site of wounded tissues [152]. Many tumorous tissues have shown to have enhanced proinflammatory cytokine production and enhanced NF-κB activity [153]. Constitutive NF-κB expression activates the gene transcription that causes proliferation, cell survival, and genomic instability, which aids in oncogenic alterations. There is evidence to suggest that inducible NF-κB pathway activation stimulates cell growth by targeting the production of cyclin D1 and prevents cell death by targeting the synthesis of BCl2 [154]. Following DNA damage, NFkB may be triggered. Acute and ongoing inflammation is brought on by NF-κB activation, which is associated with genomic instability and genetic alterations in the development and spread of cancer [155]. As a consequence, it is thought that the NF-kB signaling cascade is crucial to the pathophysiology and tumorigenesis of microbial infection. For instance, *H. pylori* infection is linked to a higher chance of developing stomach adenocarcinoma. The Correa cascade of multistep gastric carcinogenesis, also known as chronic *H. pylori* infection, is a series of events that includes metaplasia, dysplasia, and, eventually, gastric cancer [156]. A growing body of evidence from clinical follow-up research indicates that eliminating *H. pylori* greatly lowers the chance of developing stomach cancer [157,158].

## 5. Vaccination Strategies Associated

### 5.1. Human Papilloma Virus

Standard techniques for the development of vaccines for high-risk HPV infections are not feasible. Two vaccines that have received approval for the prevention of cervical cancer and HPV infections are based on virus-like particles (VLP) [159]. Large-scale clinical studies have shown type-specific protection against HPV infection and its associated cervical, vulvar, and vaginal diseases in women who were immunized prior to infection. However, in women who are already infected, little to no therapeutic effect has been observed.

### 5.2. Hepatitis Virus B and C

The surface protein (HBsAg) of the viral capsid, known as the “Australia Antigen”, was first extracted from boiling plasma to create vaccines to prevent acute HBV illness. Vaccination against HBV infection was included in the childhood vaccine initiative, and the recombinant vaccine was given FDA approval in 1986 [160]. Vaccines such as protein/peptide vaccines, antigen-antibody-based vaccines, cell-based vaccines, and genetic vaccines are the four therapeutic vaccines used to protect against hepatitis B viral infection currently being tested in clinical trials at different stages [161].

The creation of vaccines was explored for a long time, but was hampered by genetic diversity and a lack of workable animal models. In Phase I of a clinical study, CIGB-230, the first DNA therapy vaccine against HCV, was examined. In addition, with the recombinant protein of the central protein Co. 120, the vaccine comprises the plasmid pIDKE2, which is made up of the central protein (C), E1 and E2 expression signal sequence [162]. The World Health Assembly stated in 2016 that it will fight viral hepatitis and eradicate HBV and HCV by 2030. HCV elimination is now a realistic expectation thanks to the development of direct-acting antiviral (DAA) medicines in 2013, although there are still a number of obstacles in the way of HBV elimination [163].

### 5.3. Merkel Cell Polyomavirus

At present, there are currently no vaccinations or treatments available to prevent MCV infection. Adjuvant radiation and early stage local surgical excision are the main forms of treatment for infected tumors [164]. An effective method to stop viral infection is through the development of MCV vaccines. In order to suggest a fresh vaccine architecture for experimentalists to evaluate the theoretical vaccine in combating the disease, computational vaccine designing may provide an interesting alternative platform [165]. In a recent study, a chimeric-epitopes vaccine intended to protect against the virus was developed, and it stimulated strong immune responses and harbored potential B and T cell epitopes, which can be used in a chimeric multi-epitopes-based vaccine design. The vaccine was suggested as a promising candidate for the treatment of MCV and needs to be experimentally validated to initiate vaccine development [166].

### 5.4. EBV

An EBV prophylactic vaccine holds a great promise for preventative and cost-effective treatment strategies for cancers caused by EBV infections. EBV targets B cells and epithelial cells and requires multiple envelop proteins for cell entry. EBV infection of B cells and epithelial cells requires gp350, gH, gL, gb, gp42 and BMFR2, gH, gL, gB, respectively. Therefore, these proteins are the excellent targets of EBV prophylactic vaccines [167]. Virus-like particle (VLP) vaccines for EBV were created via the deletion of the EBV terminal repeats and potential oncogenes, namely EBNA2, 3A, 3B, 3C, LMP1, and BZLF1 [168]. After immunization, the EBV VLPs decreased EBV loads in the peripheral blood of humanized mice by stimulating powerful CD4+ T cell responses against structural as well as latent EBV protein epitopes [169]. Synthesized mRNA EBV vaccine development has been prompted by the quick and effective development of SARS-CoV2 vaccinations for the prevention of COVID-19. Moderna has started a phase I clinical study for its mRNA EBV vaccine after showing that their mRNA vaccine candidate, which encodes gp350, gB, gH/gL, and gp42, significantly increased EBV neutralizing titers in mice when compared to human sera. However, this approach could lead to some difficulties, such as the adverse side effects of mRNA acting as an intrinsic adjuvant that may restrict the use of multicomponent formulations and, to sustain high titers of neutralizing antibodies and powerful T cell immune responses, mRNA vaccines may require periodic booster immunizations. It is unknown if mRNA vaccinations may cause long-term memory responses in people. For infections such as EBV, which seldom cause a medical emergency, this would be unacceptable [167].

### 5.5. Kaposi’s Sarcoma-Associated Herpesvirus

Recently, the efficacy of KSHV K8.1, gB, and gH/gL as vaccine candidates was tested. Vaccinated mice and rabbits produced KSHV-neutralizing antibodies that protected epithelial, endothelial fibroblast, and B cells against infection in vitro. The creation of a KSHV vaccine may benefit from studying conserved glycoprotein-based vaccines for other herpesviruses [170].

### 5.6. Human T Cell Leukemia Virus Type 1

The idea of immunizing carriers with the aim of halting the development of ATL and HAM/TSP (HTLV-1-associated myelopathy/tropical spastic paraparesis) seems plausible. The Tax protein would be the most likely target, and a multiepitope Tax peptide vaccine has shown some preclinical promise. However, because ATL development typically takes decades, it is unclear whether a short-term surrogate end point could be used to assess vaccine efficacy—aside from perhaps showing a decrease in the level of provirus. However, the prospective benefit of this strategy for these individuals is constrained by the lack of viral gene expression in around 50% of ATL cases [170]. When the proper adjuvant was employed, a study utilizing a rat model showed that a vaccination containing the Tax peptide, which corresponds to the main epitope of CD8+ Tax-specific CTL (Cytotoxic T lymphocytes), caused the eradication of tumors. In this study, they decided to use autologous monocyte-derived DC (dendritic cells) developed in vitro as an adjuvant for the anti-ATL vaccination to help patients with ATL overcome their severe immunosuppression. Tax peptides matching to the main Tax-specific CTL epitopes previously discovered in post-HSCT ATL patients were pumped into these DC. In pilot research, three ATL patients who had responded well to earlier therapy but were unable to undergo allo-HSCT for a variety of reasons received the Tax-DC vaccination [171]. Following three injections of the Tax-DC vaccine, these patients clearly displayed Tax-specific CTL activation without experiencing any serious side effects. Two out of three patients lived for more than 4 years after receiving the immunization, according to a long-term clinical follow-up [172]. Prior to an exacerbation with lymph node swelling that was easily managed by chemotherapy, Patient 1 had maintained PR without additional therapy for 4 years. However, this patient later developed recurrent infections and passed away five years after the vaccination. Patient 2 initially had a stable illness with better performance levels, but, at 6 months, they experienced a recurrence with lymphoma without viral expression and passed away 23 months after vaccination, indicating that ATL clone-lacking Tax expression evaded the host immunity. Patient 3 is still alive after more than 5 years since receiving the immunization. Currently, three more ATL patients who received the Tax-DC vaccine using the same regimen as the pilot study have maintained CR (complete remission) for at least two years. This is part of a phase I clinical trial for the Tax-DC vaccine. The clinical results of these trials suggest that Tax-specific CTL may significantly contribute to the maintenance of remission in ATL patients until Tax-negative ATL clones appear during relapse, even if the efficacy of the Tax-DC vaccine has not yet been proven by a phase II investigation [173].

## 6. Conclusions

The evaluation of the impact of infections needs to change in light of the discovery of additional viral interactions as well as other non-viral interactions. According to previous information on the incidence of onco-pathogens in this study, there has been a substantial increase in the incidence of an association between cancer and infectious viruses. There is also a significant possibility of lowering overall cancer incidence caused by infections. Understanding the interconnections of human cancer-associated viruses, as well as other environmental drivers of pathogenicity, constitutes an increasingly significant obstacle in onco-pathogen research. We may be able to uncover crucial components in the growth and progression of cancer by focusing on exceptionally severe outcomes of relatively frequent infectious disorders and then modifying patient treatment and therapeutic cancer intervention efforts. Screening for infections in cancer cells might lead to a better knowledge of how to optimize cancer control measures. Since viral infections are more persistent than bacterial infections, viruses have a larger influence in cancer development. Thus, finding better treatment strategies with updated knowledge on pathogen-associated carcinogenesis is crucial.

## Figures and Tables

**Figure 1 pathogens-12-00770-f001:**
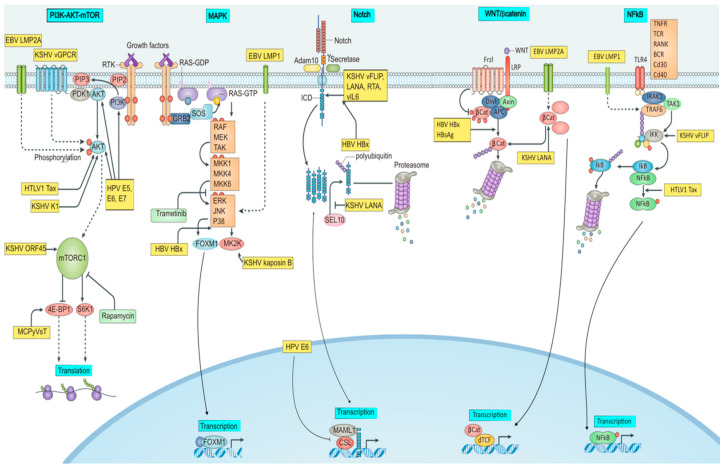
Signaling pathways modulated by onco-pathogens. The figure illustrates PI3K/AKT/mTOR, MAPK, Notch, Wnt, and NFkB signaling targeted by oncogenic viruses. BCR, or B cell receptor; E5, E6, and E7, or early proteins 5, 6, and 7; EBV, or Epstein–Barr virus; ERK1, or extracellular signal-regulated kinase 1; HBsAg, or hepatitis B surface antigen; HBV, or hepatitis B virus; HBx, or HBVX protein; HPV, or human papilloma virus; HTLV-1, or human T- Latency-associated nuclear antigen; LMP, latent membrane protein; MCPyV, or Merkel cell polyomavirus; MKK, mitogen-activated protein kinase; RAF, or RAF proto-oncogene serine/threonine-protein kinase; RTA, or replication and transcription activator; sT, or small tumor antigen; Tax, or trans activator from X-gene region; vFLIP, or viral FLICE inhibitory protein.

**Table 1 pathogens-12-00770-t001:** Pathogens and associated mechanistic actions involved in the infection-mediated cancer types.

Pathogens	Cancer Caused	Mechanism	Altered Signaling Pathways	References
Viruses	Burkitt’s lymphoma, nasopharyngeal carcinoma, Hodgkin’s lymphoma, gastric cancer	B-cell lymphoproliferations, EBNA2 and EBNA3 proteins expressed	MAPK signaling pathway	[56,57]
Epstein–Barr virus (EBV)				
Kaposi’s sarcoma virus	Kaposi’s Sarcoma, primary effusion lymphoma	Tat promotes KSHV infectivity and causes CD4+ T cells to undergo apoptosis	AKT signaling pathway, JAK/STAT signaling	[58,59,60]
Human T-cell lymphoma virus 1	Chronic, lymphoma, and acute ATLL	HBZ through increasing the noncanonical Wnt5a production	mTOR pathway signaling and Wnt pathway	[46,50]
Hepatitis virus B and C	hepatocellular carcinoma	TP53, TERT promotor, and CTNNB1 mutation by integration viral genome to host genome	p53, TERT, and WNT signaling pathways	[61,62]
Merkel cell polyomavirus	Merkel cell carcinoma	LT viral gene alterations leads to cancer development, other mutations in Rb, TP53, and PIK3CA	PI3K/AKT pathway, MAPK/ERK pathways	[63,64]
Bacterium	Esophageal adenocarcinoma, gastric cancer	Virulence factors in the bacteria and the risk factors in host leads to cancer progression	E-cadherin/β-catenin	[65,66]
*Helicobacter pylori*				
*Fusobacterium nucleatum*	Human CRC	inflammation and host immune response in the CRC microenvironment	Wnt/β-catenin	[67]
Parasites	Leukemia (Conflicting results), brain cancer	MiRNA modulation	AKT and Phosphoinositide 3-kinases (PI3Ks) pathways	[68,69,70]
*Toxoplasma gondii*				
*Schistosoma haematobium*	Bladder cancer	Eggs deposited in the urinary bladder cause irritation and tissue fibrosis initiating cancer	-	[70,71]

## Data Availability

Data are available from the authors upon request (A.V.G.).

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
