# Peer review of "Onco-Pathogen Mediated Cancer Progression and Associated Signaling Pathways in Cancer Development"

_pathogens, 2023, doi:10.3390/pathogens12060770_

Round 1

Reviewer 1 Report

The manuscript entitled "Onco-pathogen mediated cancer progression and associated signaling pathways in cancer development" clearly elaborated the major pathogens and the types of cancer caused by them. Also it mainly described the major pathways involved. overall the manuscript is written well and concise.

The manuscript is suitable for publication.

Minor comments: The authors should clearly check the typos throughout the manuscript.

Author Response

Reviewer 1 Comments: Responses

The manuscript entitled "Onco-pathogen mediated cancer progression and associated signalling pathways in cancer development" clearly elaborated the major pathogens and the types of cancer caused by them. Also, it mainly described the major pathways involved. overall, the manuscript is written well and concise.

The manuscript is suitable for publication.

Minor comments: The authors should clearly check the typos throughout the manuscript.

Response:

We are very much thankful to the reviewer for reviewing this manuscript and providing such valuable comments to improve the quality of the manuscript. We have again carefully revised the entire manuscript and have tried to address all the issues very carefully.

The manuscript has been thoroughly read and all the grammatical errors and typos has been corrected.

Author Response

 Reviewer 2 Comments: Response

We are very much thankful to the reviewer for reviewing this manuscript and providing such valuable comments to improve the quality of the manuscript. We have again carefully revised the entire manuscript and have tried to address all the issues very carefully.

In the review the authors describe the main types of cancers caused by pathogens

(Particularly by viruses) and focus on the most involved signalling pathways.

Major points:

  1. As done for oncogenic viruses, I suggest the authors to deeply describe what is known in literature about the cancer-inducing mechanisms for the cited bacteria such as Fusobacterium nucleatum, Bacteriodes fragilis, Streptococcus gallolyticus, etc, by writing individual paragraph about them.

Response: Individual paragraphs about each type of bacteria and its association in inducing cancer has been discussed under the sections 3.3 Oncogenic bacterium.

Minor points:

  1. I advise the authors to pay more attention to the English style and grammar of some phrases, hose concepts result not very clear. In light of this, I suggest the authors to revise the manuscript by making them more understandable. Here I indicate some points of the manuscript to revise:

The manuscript has been thoroughly checked again for grammar mistakes and other errors. The manuscript has been revised according to the reviewers’ comments.

Abstract:

  1. Lane 32: Understanding in this field This knowledge will give…

The statement has been revised to ‘’Gaining knowledge in this field will give important suggestions for effective pathogen-drive cancer care, control, and, ultimately, prevention.’’

Introduction:

  1. Lane 40-41: Cancer is a universally increasing health crisis combined with genetics, unhealthy routines and now infections by pathogens also promote cancer progress.

Response: This sentence has been removed and modified to ‘’ Cancer is one of the leading causes of mortality globally.’’

  1. Lane 43: after human malignancies…A reference is needed

Response: Lane 49: Reference [1] is added.

  1. Lane 44-45: Changes brought upon by this infection includes chronic inflammation

and other changes leading to oxidative stress which could eventually resulting in the cancer initiation…a synonymous of changes is better

Response: Lane 50: The sentence has been changed to ‘’Different alterations are Changes brought upon by this infection including chronic inflammation and other modifications changes leads to oxidative stress which could eventually result in the cancer initiation.’’

  1. Lane 52-53: rephrase the sentence and check the grammar

Response: Lane 58: More studies are required for a better understanding of the pathogenesis followed by these infections.

  1. Line 59: EBV… write the full name of the virus

Response: Lane 66: Epstein-Barr virus (EBV)

  1. Line 61: write the full name of the virus

Response: Lane 69: Human T-cell Lymphotropic virus type 1 (HTLV)

  1. Line 71: Helicobacter pylori and not Helicobacter pylori. Please check similar mistakes in the whole manuscript

Response: Manuscript has been thoroughly checked for Helicobacter pylori and corrected wherever necessary.

  1. Line 71-72: these viruses have been mentioned with their full name so you can use directly their abbreviation. Please revise the manuscript and do the same

Response: Lane 78: The sentences have been updated with abbreviations.

Line 78: Hepatocellular carcinoma (HCC). In the paragraph the authors start to describe this tumour with the abbreviation HCC so it is better to indicate it in the title. Please do the same for the other cancers when it is necessary

Response: Lane 85: The title and manuscript has been updated has been updated.

  1. Line 86: does not is more formal than doesn’t

Response: Lane 93: The sentence has been modified as per the suggestion.

  1. Line 93: caused mainly due the…it is a repetition, please revise

Response: Lane 100: As per the suggestion the sentence has been modified.

  1. Line 98: the verb is absent

Response: Lane 106: The two viral oncoproteins: trans activator protein (tax) and HTLV-1 basic leucine zipper factor (HBZ), are critical in the development and progression of leukaemia.

  1. Lane 101: a double space before HBZ is present… please revise the manuscript and

correct similar oversights

Response: Corrections has been done throughout the manuscript.

  1. Lane 132: 3 to 5 % of HPV infection of the cervix… it sounds better

Response: Lane 140: The sentence has been modified to ‘’It was observed that around 3 to 5 % of infection of HPV infections occurring in the of the cervix results in the transforming infection based on the cell origin.’’

  1. Line 165: consists of human viruses

Responses: Lane 172: The sentence has been modified to ‘’Several diseases including periodontal disease, HIV, UTIs and inflammatory bowel disease have already been linked to modifications or additions to the human virome, which are mostly associated with mostly consists of human viruses and bacteriophages.’’

  1. Line 182: after lymphoma a dot should be inserted and continue It is an encapsulated

virus…

Response: Lane 191: Necessary changes according to the reviewer suggestion has been made.

  1. Line 185: Alterations is plural so correct the corresponding verb. Please check all the

manuscript and correct similar mistakes

Response: Corrections has been made.

  1. Line 215: the author maybe wanted to say including and not includes

Response: Lane 225: The sentence has been modified.

  1. Line 216: I think that will is not necessary

Response: Lane 226: Th word ‘’will’’ has been removed.

  1. Line 219: for full carcinogenesis to occur… please write better this expression

Response: Lane 229: The sentence has been modified to ‘’ In experimental settings, KSHV oncogenic proteins have been found to inhibit apoptosis; however, for the establishment of carcinogenesis, additional co-factors must be present, such as co-infection with the HIV virus or the host's consumption of immunosuppressive medications.’’

  1. Lines 246-247: produced by this virus can be eliminated

Response: Lane 256: The phrase has been removed.

  1. Line 266: causes cervical cancer through the activation of oncogenes E6 and E7… it sounds better

Response: Lane 276: The sentence has been modified.

  1. Line 276-277: Hepatitis when left untreated he ongoing inflammation and liver damage cause cirrhosis… please rephrase the sentence better

Response: Lane 287: The sentence was modified to ‘’When hepatitis is when left untreated, the ongoing inflammation and liver damage cause cirrhosis, which in turn results in HCC.’’

  1. Line 284: DAAs…full name?

Response: Lane 295: Direct antiviral agents (DAAs)

  1. Line 308: Oncogenic bacteria

Response: Lane 322: The sentence has been modified with ‘’oncogenic bacteria’’ instead of ‘’carcinogenic’’.

  1. Line 314: after ulcer illness…a dot is needed

Response: Lane 325: The sentence has been modified.

  1. Line 437: after encodes…I would add and

Response: Lane 475: The sentence has been updated.